# Automatic Generation and Evaluation of Chinese Classical Poetry with Attention-Based Deep Neural Network

Jianli Zhao [1,2] and Hyo Jong Lee [2,*]

1   School of Information Engineering, Hebei GEO University, Shijiazhuang 050031, China; zhaojl@jbnu.ac.kr
2   Division of Computer Science and Engineering, CAIIT, Jeonbuk National University, Jeonju 54896, Korea
*   Correspondence: hlee@jbnu.ac.kr; Tel.: +86-063-270-240

**Abstract:** The computer generation of poetry has been studied for more than a decade. Generating poetry on a human level is still a great challenge for the computer-generation process. We present a novel Transformer-XL based on a classical Chinese poetry model that employs a multi-head self-attention mechanism to capture the deeper multiple relationships among Chinese characters. Furthermore, we utilized the segment-level recurrence mechanism to learn longer-term dependency and overcome the context fragmentation problem. To automatically assess the quality of the generated poems, we also built a novel automatic evaluation model that contains a BERT-based module for checking the fluency of sentences and a tone-checker module to evaluate the tone pattern of poems. The poems generated using our model obtained an average score of 9.7 for fluency and 10.0 for tone pattern. Moreover, we visualized the attention mechanism, and it showed that our model learned the tone-pattern rules. All experiment results demonstrate that our poetry generation model can generate high-quality poems.

**Keywords:** classical Chinese poetry; deep learning; transformer-XL; self-attention; BERT-CCPoem





## 1. Introduction

The first collection of classical Chinese poems, known as the Book of Songs, was compiled by Confucius over 2500 years ago. By the time of the Tang Dynasty (618 to 907 AD), numerous brilliant and timeless classical poems had been composed. Meanwhile, the regulated verse and quatrain had become the most popular forms of classical poetry [1]. Poems with eight sentences, regulated by tone and rhyming-rules, are known as the regulated verse. In addition, each sentence consists of five or seven Chinese characters. The quatrain is akin to the regulated verse but only comprises four sentences, such as the example of a 5-character quatrain shown in Figure 1.

To ensure poetry has a harmonic melody, four tonal patterns are created in the regulated verse and quatrain, such as alternative-rule, opposite-rule, alike-rule, and rhyming-rule. Firstly, the tones of traditional Chinese characters are divided into two categories of poetry rhythm: the Ping-tone (level tone) and Ze-tone (oblique tone). Secondly, for the alternative-rule, the Ping-tone and Ze-tone exist alternately at the 2nd, 4th, and 6th positions in a sentence of 7-character poetry, and only at the 2nd and 4th positions in 5-character poetry. The opposite-rule states that in a pair of sentences, the tone of the next sentence in the 2nd, 4th, and 6th positions must be contrary to that of the previous sentence in the corresponding locations. The alike-rule is a rule that applies to the sentences in two adjacent pairs. At the 2nd, 4th, and 6th positions, the tone of the first sentence in the next pair must be the same as the tone of the second sentence in the previous pair. To make the poem sound better, the rhyming-rule requires the rhymed last position of even lines with the Ping-tone type of tones, and simultaneously, odd lines have to remain unrhymed, except for the first line.

相思

| Line | Title: Missing the Beloved |
|---|---|

1  红豆生南国，                (P Ⓩ P Ⓟ Z)

Red beans growing in the southern countryside,

2  春来发几枝 (zhi)。          (P Ⓟ P Ⓩ P̲)

Spring hangs them on new branches of height.

3  愿君多采撷，                (Z Ⓟ P Ⓩ Z)

Wish you could pick as many as possible,

4  此物最相思 (si)。            (Z Ⓩ Z Ⓟ P̲)

Because they would send the beloved to your side.

**Figure 1.** An example of a 5-character quatrain where P and Z represent Ping-tone and Ze-tone, respectively. The circled P and Z denote the key positions that must obey the basic tone rules. The underscored Ps represent these positions that must be rhymed.

Classical Chinese poetry not only boasts rich imagery and a lovely cadence, but it also preserves Chinese culture and history. Therefore, automatic poetry generation is an interesting and challenging research topic. The study of automatic poetry generation is beneficial for constrained language generation and ancient Chinese text comprehension tasks. It can also assist poets in creating poems, and contribute to the teaching process and research of ancient poetry.

Automatic poetry generation has attracted the attention of many researchers in recent years. The template-based method masks some words of an existing poem as template, and then replaces them with some other words to produce a new poem. The poetry generated by this method has been improved in grammar, but the flexibility is too poor. The statistical approach to generating classical Chinese poetry [2] has been proposed, which takes certain keywords as input and generates sentences one by one to form a poem. However, these traditional methods rely to a high degree on professional knowledge in poetry and, therefore, require experts to design many manual rules to constrain the rhythm and quality of the generated poetry.

Deep learning methods have made great progress in machine translation and text generation. In classical Chinese poetry generation, a language model based on a recursive neural network (RNN) [3,4] can learn poetry's structure, semantics, and coherent constraints without additional manual rules concerning limitations of rhyming and tone. However, the RNN-based model has difficulties in generating long sentences because the gradient vanishing problem restricts the RNN to operate only on a short-term memory. Attention Based Sequence-to-Sequence Model [5,6] was also introduced to study classical Chinese poetry generation. The attention and long-short term memory (LSTM) mechanism [7] in sequence-to-sequence models facilitates semantic consistency throughout the generated poetry, but it is still difficult for the model to catch effective long-term relationships because the context information of the RNN encoder decays as the time step increases.

An important problem is that the above methods only focus on generating the quatrains. Since RNNs inherently lack the ability to capture long sequences, these RNN-based models cannot effectively compose longer poetry, such as the regulated verse. Therefore, we propose a novel model for the automatic generation of poetry that can generate longer poems with high consistency. The model learns the deep semantic representation of Chinese characters in context and uses the multi-head attention mechanism [8,9] to capture various relationships between the sentences of poems.

Another challenging task is the automatic evaluation of the quality of poetry. The common approaches for assessing the quality of poems rely on human experts or some automated metrics such as the Bilingual Evaluation Understudy (BLEU) method [10]

and the Metric for Evaluation of Translation with Explicit Ordering (METEOR) [11]. The evaluation of the quality of numerous poems by human experts is time-consuming and expensive. BLEU and METEOR compute the n-gram overlap between generated sentences and references, mainly for tasks such as machine translation, but for tasks such as poetry generation and dialogue response generation, these metrics have little correlation with human judgments [12]. To solve this problem, we propose an automatic evaluation model for classical Chinese poetry that includes two modules: the tone-pattern-checker and text-fluency-checker. The tone-pattern-checker uses the formerly mentioned rules of rhythmic poems to calculate the score of poems. The text-fluency-checker module uses a pre-trained BERT model based on the classical Chinese poetry corpus to check the fluency of the poem and the consistency among the sentences, and finally gives a score.

In summary, the main contributions are three-fold:

(1) We propose a novel classical Chinese poetry generation model based on Transformer-XL, which learns the deep representative of Chinese characters and captures longer-term dependencies of Chinese poetry.
(2) We develop a novel automatic evaluation model for classical Chinese poetry that consists of a fluency checking module and a tone-pattern checking module.
(3) We visualize the attention mechanism in our model and show that our model fully captures the tone pattern of poems.

The rest of the paper is organized as follows: in Section 2, we review the related works of classical Chinese poetry composition. In Section 3, we state the classical Chinese poetry generation model and the automatic evaluation model. We summarize the experimental results and the analysis of our poetry model in Section 4. Finally, we provide a brief conclusion in Section 5.

## 2. Related Works

The statistical machine translation-based poetry generation [2] learns tone and rhyming from a line that is regarded as a source sentence. Then, it translates the previous line into the next line, which is regarded as a target sentence by the machine translating model. Statistical methods of poetry generation require additional expert rules to constrain the rhythm of poetry.

However, RNN-based sequence-to-sequence models [13–19] of Chinese poetry generation can capture poetic content and form with no additional manual rules. A Chinese poetry generation model based on the RNN [13] was presented, in which the poem lines were composed incrementally, taking into account everything that had been generated so far. A new generative model was proposed by Yan [14] that adopted a polishing scheme to generate one poem incrementally and iteratively by refining each line. A conditional variational autoencoder (VAE) in [17,18] is employed to enhance the thematic consistency of generated verses with the title. Yang et al. [19] proposed a novel poetry generation model that employs a sequence-to-sequence model with attention mechanisms and incorporates the mutual information to strengthen the relationship between manually selected style inputs and generated style-specific outputs. Since RNNs inherently lack the ability to capture long sequences, the above RNN-based approaches only focus on the generation of a quatrain and cannot effectively generate an eight-line regulated verse with high consistency.

Some text generation models based on a generative adversarial network (GAN) [20,21] also explored the generation of classical Chinese poetry. The above-mentioned models can compose quatrains without any prior knowledge of special structure rules in poetry, such as tonal patterns rules used to create quatrains. Nevertheless, they only investigated the generation of quatrains due to lacking a deep understanding of the semantics of poetry. To address the above issue, we present a novel poetry generation model that utilizes the pure self-attention mechanism to capture the deep semantics of Chinese characters and adopts the segment level recurrence mechanism to learn the longer-dependency among sentences in poems.

## 3. Methods

The Chinese language is usually expressed as a sequence of Chinese characters, and it can be abstracted into a probabilistic model that maximizes the joint probability of each legal sentence in the language. The probability of a sequence $(c_1, c_2, \ldots, c_T)$ can be commonly factorized as the product of conditional probability.

$$P(c_1, c_2, \ldots, c_T) = \prod_{t=1}^{T} P(c_t | c_1, c_2, \ldots, c_{t-1}), \tag{1}$$

The goal of language modeling is to learn the distribution function of sentences in the language.

The representation of word vectors is crucial for language models. In the early one-hot vector method, word vectors are independent and cannot reflect the correlation between words. Word embedding, such as Skip-gram [22] and Glove [23], represent the semantic information of words by word vectors in an embedding space, where semantically similar words are close to each other in the space. However, the word vectors learned by this method are fixed, which is obviously not suitable for polysemous words. ELMo [24] solved the word sense disambiguation problem by learning deep contextualized word representations, but limited by LSTM, it cannot capture long-term dependencies. By adopting a multi-head self-attention mechanism, the transformer learns dynamic word embedding and captures long-term dependencies. Transformer-based models, such as Transformer-XL, GPT [25], and BERT [26], have achieved state-of-the-art results in many NLP tasks. Therefore, we propose a poetry automatic generation model using Transformer-XL and a BERT-based poetry automatic evaluation model.

### 3.1. Transformer-Xl Model

To capture longer-dependencies, Transformer-XL is presented, which combines the advantages of RNN sequence modeling with the transformer self-attention mechanism [27–30]. On each segment of the input data, it adopts the transformer's attention module and employs a recurrent mechanism to learn the dependencies between successive segments. The dependency that Transformer-XL learned is 80% longer than RNNs and 450% longer than vanilla transformers [31], resulting in superior performance on both short and long sequences [8].

Transformer-XL adopts the self-attention with the multi-heads mechanism of the vanilla transformer, but the difference is that it adopts a more simplified and efficient framework with only a decoder, instead of the encoder-decoder framework. The self-attention mechanism captures the dependencies between words within a sentence, whereas the traditional attention mechanism captures the dependence between words in source sentences and words in target sentences.

The vanilla transformer needs to be fragmented when processing long text sequences, which causes the semantic segmentation problem. The transformer-XL introduces a recurrence mechanism that recursively reuses the hidden state of the previous segment when calculating the hidden state of the current segment. Thus, the Transformer-XL model can obtain deeper dependencies and more abundant representations than the vanilla transformer.

### 3.2. Classical Chinese Poetry Generation Model

We exploit the power of capturing the longer-term dependency of Transformer-XL to construct the classical Chinese poetry generation model (CCPTXL) [32] in Figure 2 in three steps. The transformer block consists of 16 layers and each layer adopts a multi-head mechanism with 10 heads. The classical Chinese poetry generation model can output poems according to the input Chinese characters as shown in Figure 2a.

Firstly, the model uses an embedding approach to convert the input sequence into vectors, and then it combines the embedding vectors with relative positional encodings.

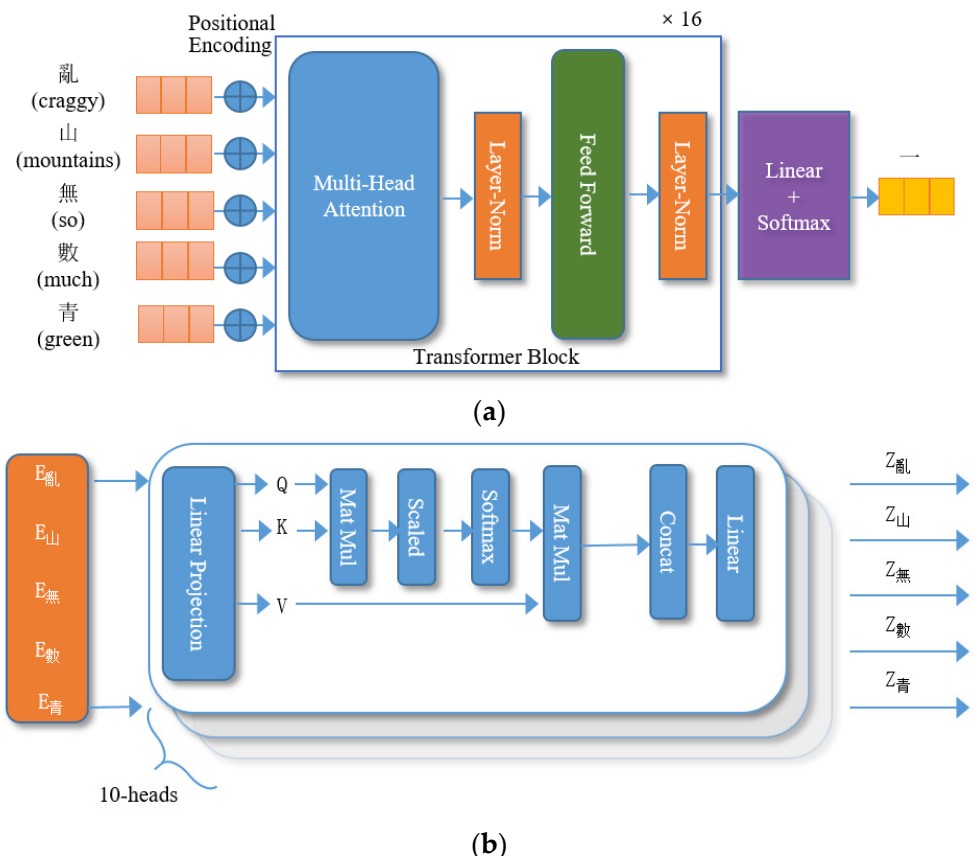

**Figure 2.** The classical Chinese poetry generation model based on Transformer-XL (CCPTXL): (**a**) The classical Chinese poetry generation model and (**b**) Multi-Head Attention Layer.

Secondly, the transformer block's multi-head attention module calculates the attention scores of the input sequence. As we all know, any two Chinese characters can have various relationships, including meaning, part-of-speech, form, tone, rhyme, and more. We use a multi-head attention module with ten heads to learn these relationships. The input embeddings are mapped into different sub-spaces such as Q, K, V, and then the attention scores are calculated by Equations (2) and (3). We utilize the Layer-Normalization module to normalize the optimization space to speed up model convergence. The vectors from the previous module are then projected to a higher dimensional space by the Feed-Forward module, making it easier to discern between the various forms of information in the vectors.

$$\text{Attention}_{\text{Score}} = \text{softmax}\left(\frac{QK^T}{\sqrt{d_k}}\right)V \tag{2}$$

$$\text{MultiHead}_{\text{Attention}} = \text{Concat}(\,head_1,\,head_2,\ldots,\,head_h)W^O \tag{3}$$

Finally, we employ a fully connected layer to project the vectors from the 16 stacked Transformer Blocks into a larger vocabulary vector space. Then the following softmax module calculates each character's probability and selects the character with the highest probability as the output of each time step.

### 3.3. Automatic Evaluation Model Based on BERT-CCPoem

Automatically evaluating the quality of generated poems is also a very challenging issue. Human experts [2,12–19] or the automatic metrics BLEU technique [2,13,14,16–18,20,21] are the most prevalent approaches for assessing the quality of poems. The evaluation of the quality of a large number of poems by human experts is time-consuming and expensive. Moreover, different experts have their appreciation preferences for poems, so it inevitably

leads to a certain subjectivity. BLEU is an automated and inexpensive measure standard for machine translation that achieves a high correlation with human judgments of quality, but poetry generation is fundamentally different from machine translation. First, the source and target sentences in machine translation attempt to have the same meaning, but the poetry creation is different. The target sentence does not repeat the meaning of the source sentence, although it is based on the source sentence. Second, BLEU is a method based on word mechanical matching, which cannot compute the sentences relationship according to the semantic meaning of words. Thus, both tone-pattern-checker and text-fluency-checker are important in an automatic poetry evaluation model. The tone-pattern-checker can give a score after identifying whether a poem follows the tone and rhyming-rules of poetry. The algorithm of a tone-pattern-checker is shown in Algorithm 1, and consists of structure detection and tone rhyming detection. The structure detection is to determine whether a poem follows the format of the 5-character quatrain, 7-character quatrain, 5-character regulated verse, or 7-character regulated verse. The tone rhyming detection is to detect whether a poem obeys the alternative-rule, opposite-rule, alike-rule, and rhyming-rule, and finally gives a corresponding score.

---

**Algorithm 1** Tone-Pattern-Checker

---

1: Input: one quatrain or regulated verse noted as X, tone_score = 0;
2:   if the structure of X follows the rules of quatrain or regulated verse then
3:       tone_score = 2
4: end if
5: if X follows the alternative-rule then
6:    tone_score = tone_score + 2
7: end if
8: if X follows the opposite-rule,
9:    tone_score = tone_score + 2;
10: end if
11: if X follows the alike-rule then
12:     tone_score = tone_score + 2
13: end if
14: if X follows the rhyming-rule then
15:     tone_score = tone_score + 2
16: end if
17: return tone_score.

---

The text-fluency-checker is responsible for detecting whether a poem follows ancient Chinese grammatical norms, whether it reads fluently, whether the collocation of words is reasonable, and whether the sentences are coherent. Moreover, it gives corresponding quantitative scores to poems.

By using the deep representations of ancient Chinese provided by the pre-trained BERT-CCPoem [33], we model poem fluency evaluation as a text classification problem, as shown in Figure 3. The BERT model is a transformer-based bidirectional mask language model that has achieved SOTA results on many natural language processing tasks, including GLUE, SQuADv1.1, SQuADv2.0, and others. BERT-CCPoem is a pre-trained BERT-based model for classical Chinese poetry by the Research Center for Natural Language Processing of Tsinghua University. BERT-CCPoem learns plentiful semantic and grammatical information on Chinese characters because it is trained on a large amount of the Chinese corpus. Therefore, using the semantic vectors provided by BERT-CCPoem, we add a classifier over BERT-CCPoem to fine-tune a text-fluency-checker.

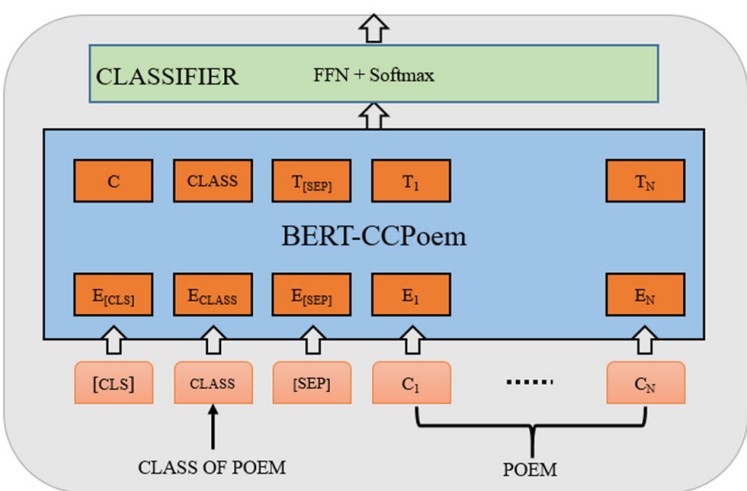

**Figure 3.** The text-fluency-checker fine-tuned on BERT-CCPoem.

## 4. Experiment and Results

### 4.1. Dataset Preparation

We collected 123,672 classical Chinese poems from the Internet as our corpus. Our corpus mainly contains regulated verse and quatrain poems with 5-character and 7-character, having all kinds of poetry styles, such as pastoral, descriptive, romantic, and more. We randomly chose 80% of our corpus as a training set, 10% as a validation set, and 10% as a test set.

### 4.2. Training

We detail the CCPTXL model structure and parameters in Table 1. The vocabulary size of our corpus is 10,091. The dimension of word embedding vectors was set to 410. We used 41-dimensional vectors Q, K, and V in the attention layer. We had 16 identical transformer-blocks and employed 10 heads of attention, as shown in Figure 2. The total number of trainable parameters was 46M. In order to better converge to the global optimum, we trained our model using Adam optimization and dynamic learning rate policy. The initial learning rate was 0.0001, which then decayed using the cosine schedule. To prevent the model from overfitting, the dropout mechanism was used in the Multi-head layer and Feed Forward layer, and the dropout rate was set to 0.1.

**Table 1.** The model structure and parameters.

| Layers | Output Size | Trainable Parameters |
|---|---|---|
| Embedding layer | (100, 64, 410) | 4,137,310 |
| Multi-head layer [1] | (100, 64, 410) | 10,758,400 |
| Feed Forward layer [1] | (100, 64, 410) | 27,552,000 |
| Linear_softmax layer | (100, 64, 10,091) | 4,137,310 |
| The total number of trainable parameters | | 46,585,020 |

[1] There are 16 identical Multi-head layer and Feed Forward layers.

### 4.3. Evaluation for Generated Poems

In order to fine-tune a text-fluency-checker, we first needed to construct positive and negative data sets. It was easy to build the positive data set by directly extracting some poems from the classical Chinese poems' dataset, whereas it was hard to create a negative data set of numerous poems written by humans. Therefore, we introduced noise into the manual poetry, which caused the original correct word order to be disrupted, resulting in a certain degree of grammatical errors, thereby automatically generating a large number of

negative poems. In addition, we were able to control the degree of grammatical errors in the negative data set according to the amount of noise introduced.

From the classical Chinese poems dataset, we randomly extracted 12,000 poems that consisted of 6000 5-character and 6000 7-character regulated verse poems. The selected dataset consisted of 3 levels. Level 0 was a completely correct human-made poetry dataset without noise; Level 1 introduced 30% noise; and Level 2 introduced 60% noise on the original poetry dataset. To fine-tune the BERT-CCPoem model, we use the training dataset of 3 levels, which amounts to 36,000 poems. We also generated the validation dataset and testing dataset, which amounts to 7200 and 3600 poems, correspondingly.

The text-fluency-checker gives a total score of 10, and each poem can get 10, 7, or 4 points corresponding to Level 0, Level 1, and Level 2, respectively. In the tone-checker, each poem also receives a total score of 10, with 2 points for the structure of quatrain or regulated verse, 2 points for the alternative-rule, 2 points for the opposite-rule, 2 points for the alike-rule, and 2 points for the rhyming-rule.

We generated two 1000-poems-datasets by our CCPTXL model and a classical Chinese poetry model based on sequence to sequence with attention (CCPSA), respectively. Moreover, we also randomly extracted 1000 human-made poems from the classical Chinese Poetry Dataset (CCPD). Each of the three evaluation datasets consisted of 500 5-character and 500 7-character regulated verse. Besides assessing the three poetry datasets above, we also evaluated the example quatrains in [2,12,13,15–19] using the text-fluency-checker and the tone-checker. Although the number of example poems shown above is 1, 2 or 3 at most, it still has some reference values. The average scores are shown in Table 2.

**Table 2.** The Comparison of Poems' Scores.

| Poems | Tone-Checking Score | Fluency-Checking Score |
| --- | --- | --- |
| SMT Models [2] | 7.0 | 10.0 |
| Images2Poem [12] | 9.5 | 6.5 |
| RNNPG [13] | 9.8 | 10.0 |
| Attention Model [15] | 8.5 | 7.0 |
| PG [16] | 9.0 | 10.0 |
| CVAE-HD [17] | 10.0 | 10.0 |
| CVAE-D [18] | 9.0 | 7.0 |
| SPG [19] | 10.0 | 10.0 |
| CCPSA [1] | 6.9 | 7.6 |
| CCPD [1] | 9.5 | 9.3 |
| CCPTXL (our model) [1] | **10.0** | **9.7** |

[1] The models generate regulated verses, whereas the other models only generate quatrains. Bold numbers represent the highest performance.

The average fluency-checking score from our CCPTXL model was higher than from CCPSA. Furthermore, it also obtained a higher score than these randomly chosen poems from CCPD. SPG [19] and CAE-HD [17] obtained high scores on fluency but they are only quatrains, whereas our generated regulated verses are twice as long as theirs. This demonstrates that our poem model can generate more fluent and coherent longer sentences. The distribution of the poems of each data set on the three levels is shown in Figure 4. Among the generated 1000 poems from our model, 98.3% were evaluated as Level 0, 1.7% as Leve1, and no poems were Level 2. This shows that our model can stably compose high-quality poems.

The tone-checking scores of poems were calculated by the tone-checker, in which each poem is checked and scored according to the tone and rhyming rules. Our CCPTXL model obtained the highest score of 10, as shown in in Table 2. This shows that almost all the poems generated by our model can obey the tone and rhyming rules of regulated verse. Figure 5 shows that the poems composed by our CCPTXL model scored 9–10 and 8–9, accounting for 89.5% and 7.9%, respectively. The result indicates that our model accurately captured the tone and rhyme features.

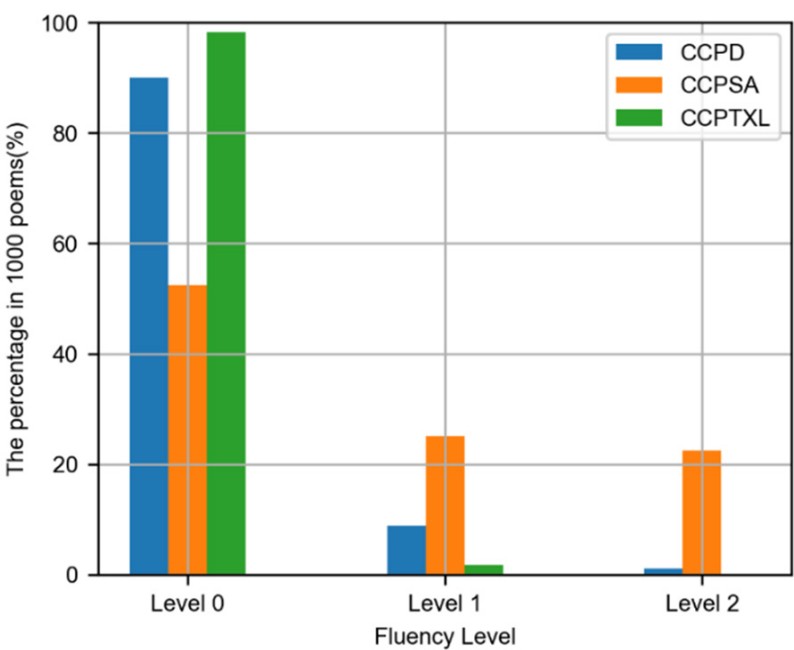

**Figure 4.** The distribution of fluency level of poem datasets.

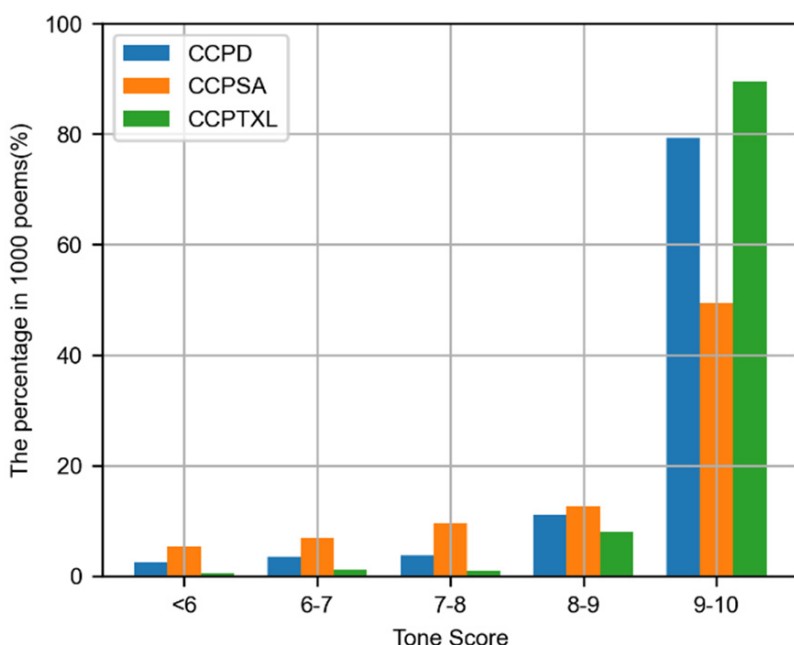

**Figure 5.** The distribution of tone score of poem datasets.

### 4.4. Analysis of the Generated Poems

Our poetry model not only completely captured the tone and rhyme rules of classical Chinese poetry, but also even some advanced patterns. There is a very typical expressive pattern of classical Chinese poetry in which a poet usually describes the objective landscape or things and then sublimes the previous description into his feeling, opinion, or philosophy at the final lines. An example of the 7-character regulated verse generated by our poetry model shown in Figure 6 has eight sentences (four pairs) with 7 characters in each sentence.

We utilized the example in Figure 6 to verify the tonal and rhyming rules that our poetry model grasped. The annotations of the tonal patterns in the right column in Figure 6 show that the generated poem fully adheres to the four basic rules. As we described earlier for the alternate-rule, the tones of the 4th and 6th characters are determined by the

2nd. When visualizing what the poetry model learned, we found strong evidence for this rule, and we further figured out which parts are responsible for the alternative-rule in the attention module with multi-layers and multi-heads.

The horizontal axis represents the last characters of the current character to be generated, and its maximum length is 100. The vertical axis represents the ten heads in a layer. We adopt a heat map method to be more intuitive, where the grey squares represent the attention scores. The darker the color of a grey square, the greater the attention score, which means that the character on the corresponding horizontal axis is more important to the generation of the current character.

Line
1       Title:書房
        Title: study room
2       一點寒燈照竹房 (fang)，                    (Z Z P P Z Z P)
        The lamp glimmering the bamboo study room at frosty night,
3       詩書教子讀南陽 (yang)。                    (P P Z Z P P P)
        I was teaching children to read the Zhuge Liang's story.
4       不因名字傳青史，                          (Z P P Z P P Z)
        Not for the sake of leaving a name in the history，
5       只恐功名負赤囊 (nang)。                    (Z Z P P Z Z P)
        I was just afraid that I will not be able to fight for our country.
6       風雨夜窗孤枕夢，                          (P Z Z P P Z Z)
        The night wind and rain hitting on the window,
7       江湖春水一篷霜 (shuang)。                  (P P P Z Z P P)
        I have ever wandered in the world and experienced hardships.
8       何由洗盡英雄氣，                          (P P Z Z P P Z)
        But still not change the heroic spirit,
9       會向明時謁帝王 (wang)。                    (Z Z P P Z Z P)
        I have been waiting for the opportunity to serve the country.

**Figure 6.** An example of regulated verse generated by our CCPTXL model.

Figure 7a shows that the 4th character lamp (燈) with the 2nd character dim (點) has the maximum attention score at head 5 in layer 4. Figure 7b also shows that at head 5 in layer 4, the largest attention score exists between the 6th character bamboo (竹) and the 4th character lamp (燈). This indicates that the 4th character strongly depends on the 2nd character and the 6th character on the 4th character. Moreover, their tonal pattern is Ze-tone, Ping-tone, and Ze-tone in order, i.e., Ze-tone and Ping-tone appear alternately. Thus, this kind of strong dependency in the same line of poetry can only be the alternative-rule. The same situation also appears in Figure 7c,d. After observing and counting many sentences with different heads at different layers, we conclude that head 5 in layer 4 completely controls the alternative-rule.

According to the opposite-rule, in a pair of sentences, such as lines 1 and 2 in Figure 6, the tone of the second character dim (點) in line 1 is Ze-tone, so the second character book (書) in line 2 must be Ping-tone. In Figure 8a, the character book (書) has a large attention score on the character dim (點) at head 6 in layer 8. Based on more observations of the same phenomenon as Figure 8, we infer that head 6 in layer 8 controls the opposite-rule in the first and last pair, but head 5 and head 6 in layer 8 catch the opposite-rule in the second and third pair.

In Figure 9a, the visualization of the alike-rule shows that the second character reason (因, Ping-tone) in line 3 must follow the same tone as the second character book (書, Ping-tone) in line 2. In Figure 9b,c, the adjacent pairs also follow the alike-rule, in which the tone of the first sentence in the next pair must be the same as the tone of the second sentence in the previous one. After many observations of generated poems, we conclude that the head 8 or head 2 in layer 10 represents the alike-rule.

For the rhyming-rule, the last characters in the even lines i.e., sun (陽, yang), bag (囊, nang), frost (霜, shuang), and monarch (王, wang) have the same rhyming and are all Ping-tone. In Figure 10a, we can find that bag (囊) has a very strong dependency on sun (陽) at head 8 in layer 15 and the same phenomenon goes for frost (霜) with a dependency on bag (囊) and monarch (王) with a dependency on frost (霜), as seen in Figure 10 b,c. In addition, monarch (王) has not only a very strong dependency on frost (霜) but also a relatively strong relationship on the more distant rhyming character bag (囊), as shown in Figure 10c.

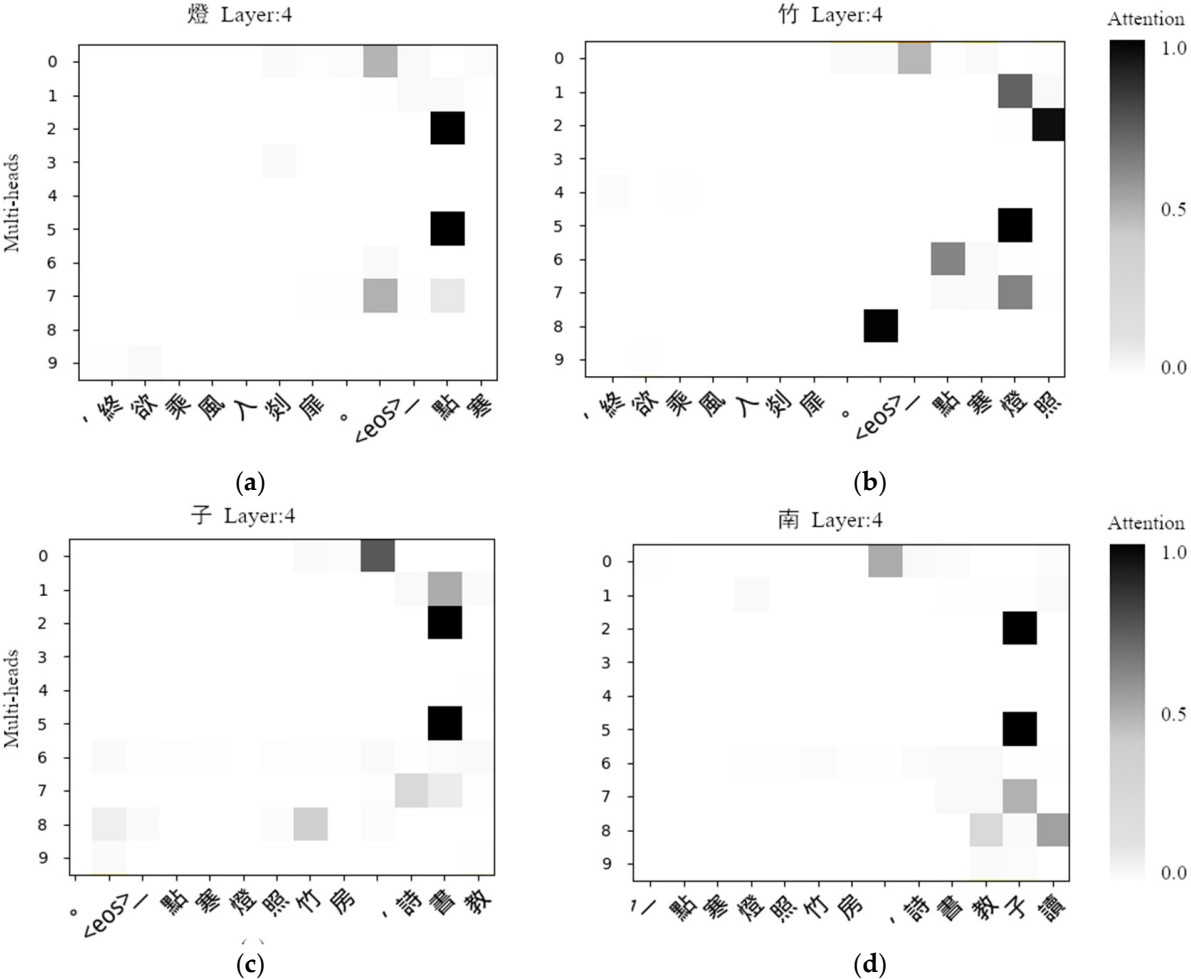

**Figure 7.** Visualization of the alternative-rule. The grey squares represent the attention score and the darker the color, the larger the score. (**a**) lamp (燈); (**b**) bamboo (竹); (**c**) son (子); (**d**) south (南).

After much observation, we confirmed that the head 8 in layer 15 controls the rhyming-rule.

Through visualizing the poetry rules, we demonstrate that our model learns the tonal patterns and rhyming rules very well. Moreover, our poetry generation model also acquired some knowledge of expression patterns that are more advanced and abstract than basic tonal rules. For example, the poem generated in Figure 6 describes some objective scenery or things in lines 1 to 6, such as the lamp and bamboo study room, book and story, history, wind, and rain, but in the last two lines, our model combines the preceding scene and expresses its emotions and views on life like a real poet. This expression pattern is a very typical way of writing ancient Chinese poetry.

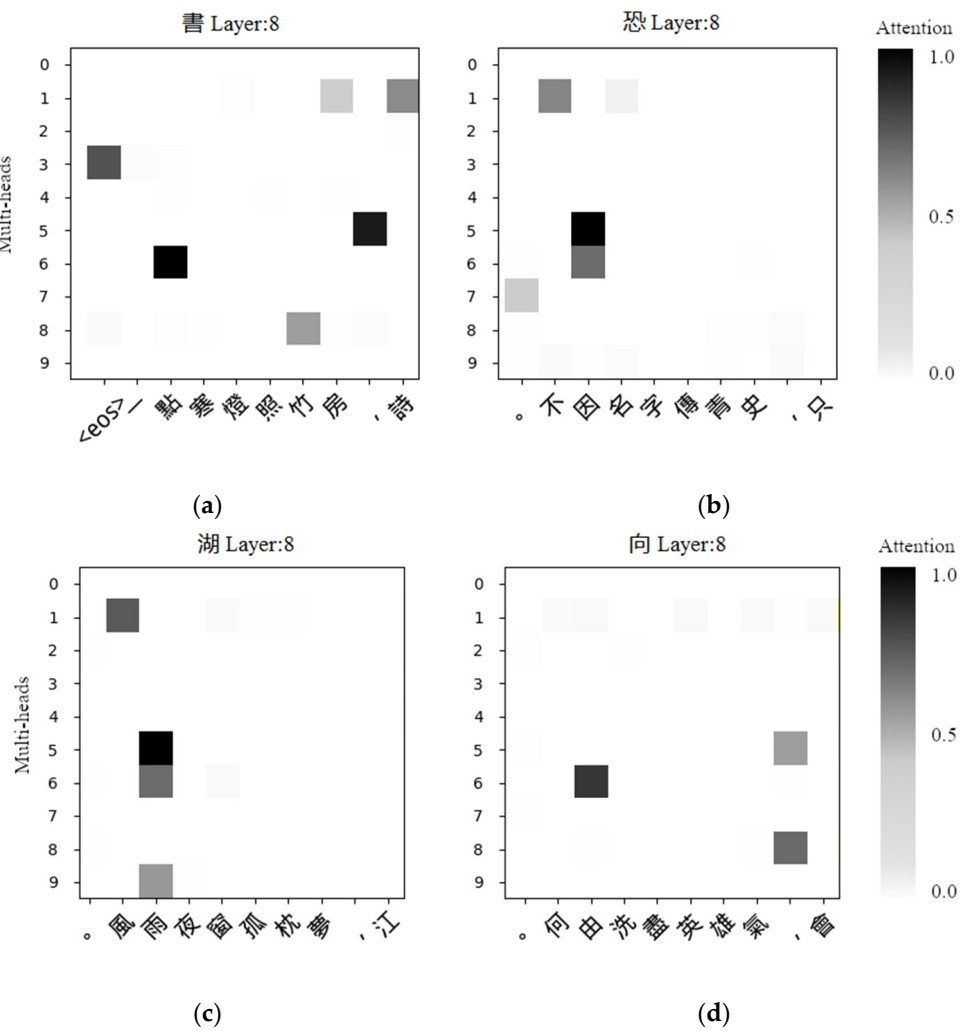

**Figure 8.** Visualization of the opposite-rule. (**a**) book (書); (**b**) afraid (恐); (**c**) lake (湖); (**d**) to (向).

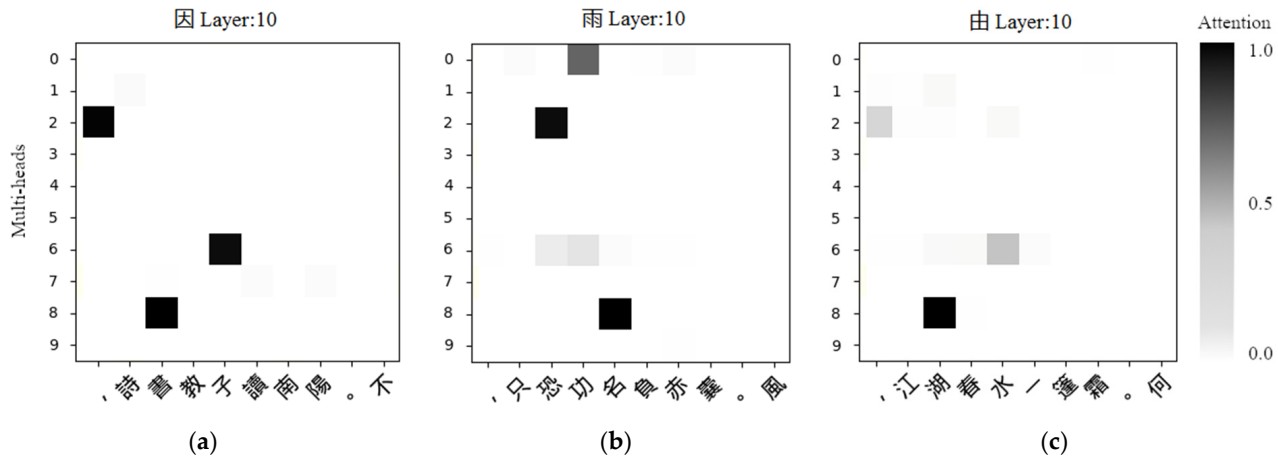

**Figure 9.** Visualization of the alike-rule. (**a**) reason (因); (**b**) rain (雨); (**c**) not (由).

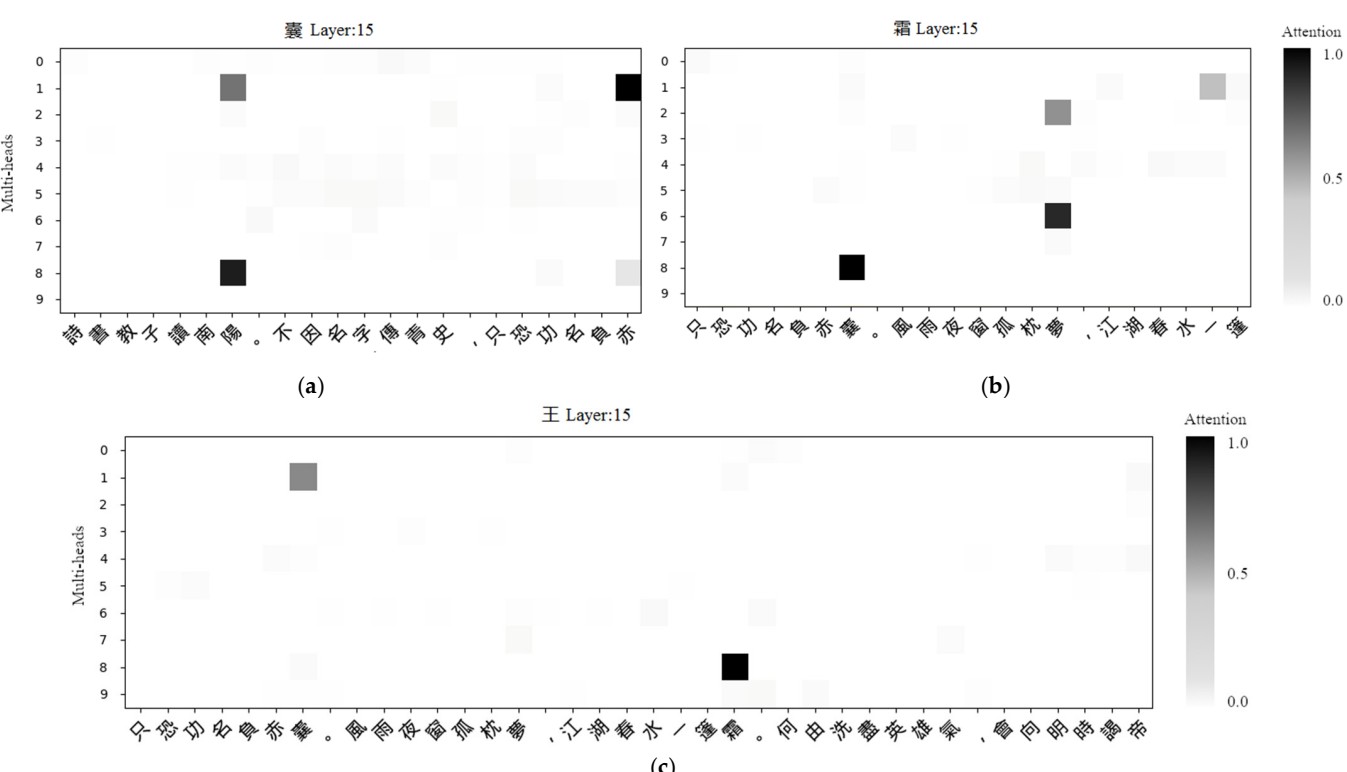

**Figure 10.** Visualization of the rhyming-rule. (**a**) bag (囊); (**b**) frost (霜); (**c**) monarch (王).

## 5. Conclusions and Future Work

In this paper, we proposed a novel classical Chinese poetry model. Benefiting from the multi-head self-attention and segment-level recurrence mechanism, our model can capture the longer-term dependency of Chinese characters. Furthermore, by visualizing our poetry model and analyzing the attention score, we prove that our poetry model fully captures the tone pattern of classical Chinese poetry. We also present a novel automatic poem evaluation model that consists of fluency checking and tone-pattern checking. The experimental results from the evaluation system show that our model obtains an average score of 9.7 for fluency and 10.0 for tone pattern. The comprehensive experiments confirmed that our poetry generation model can generate high-quality poetry. For future work, we will consider enhancing the diversity of generated poems according to the same keywords. We also plan to improve the automatic evaluation model in terms of theme consistency and meaningfulness of poetry.

**Author Contributions:** Conceptualization, J.Z. and H.J.L.; methodology, J.Z.; software, J.Z; validation, J.Z. and H.J.L.; formal analysis, J.Z.; writing—original draft preparation, J.Z.; writing—review and editing, J.Z. and H.J.L.; supervision, H.J.L.; funding acquisition, H.J.L. All authors have read and agreed to the published version of the manuscript.

**Funding:** This work was supported by the Basic Science Research Program through the National Research Foundation of Korea (NRF) funded by the Ministry of Education under Grant GR2019R1D1A3A03103736.

**Institutional Review Board Statement:** Not applicable.

**Informed Consent Statement:** Not applicable.

**Data Availability Statement:** Not applicable.

**Conflicts of Interest:** The authors declare no conflict of interest.

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
