# Peer review of "Automatic Generation and Evaluation of Chinese Classical Poetry with Attention-Based Deep Neural Network"

_applsci, doi:10.3390/app12136497_

Round 1

Reviewer 1 Report

 1.      CCPTXL model represented in Figure 2 shows the overall block diagram.  Mention the total number of layers used in attention based deep neural network model.

2.       Include a table showing the entire configuration and the parameter details at each layer. The total number of trainable and non-trainable parameters can be highlighted.

3.      In Table 1, algorithm of tone-pattern checker, how the threshold value is chosen as 2 for tone_score. 

4.      Mention all the attributes present in the dataset.

5.      How the automatic feature extraction process happens in the model.

6.      Separate block diagram for multi-head attention model can be included. 

7.      Mathematical notations are missing.  Include the related mathematical notations for all the modules.

8.      What technique is used for validating the results of the model.

9.      Mention the future scope of the work carried out.

Reviewer 2 Report

This is an article in which the authors use existing techniques, and no innovative elements are squeezed into the work.

Provide more information and technical explanations about the Transformer XL compared to other similar models (write recurrences that are used, etc.).

In general, there is no argumentation about the specific methodological choices, techniques, and parameters.

Point out syntactic and semantic differences between Chinese and English: Are there similarities? Can your method be applied in English?

What is your future work - are there plans to improve the approach?

Round 2

Reviewer 2 Report

The authors answered all my questions and accepted all my advice.

Thank you!